# Optimal Fractionation Scheduling for Radiotherapy Treatments with Reinforcement Learning, Tumor Growth Modeling and Outcome Modeling

**DOI:** 10.3390/biomedicines13061367

**Published:** 2025-06-03

**Authors:** Mélanie Ghislain, Florian Martin, Manon Dausort, Damien Dasnoy-Sumell, Ana Maria Barragan Montero, Benoît Macq

**Affiliations:** 1Institute for Information and Communication Technologies, Electronics and Applied Mathematics (ICTEAM), UCLouvain, 1348 Louvain-La-Neuve, Belgium; florian.martin@uclouvain.be (F.M.); manon.dausort@uclouvain.be (M.D.); damien.dasnoy@uclouvain.be (D.D.-S.); benoit.macq@uclouvain.be (B.M.); 2Molecular Imaging, Radiotherapy and Oncology Institute, UCLouvain, 1200 Woluwe-Saint-Lambert, Belgium; ana.barragan@uclouvain.be

**Keywords:** reinforcement learning, radiotherapy treatment planning, tumor growth simulation, cancer complication

## Abstract

**Objective:** Radiotherapy is a primary method for cancer treatment, wherein radiation doses are divided into multiple sessions or fractions to effectively target tumors and minimize damage to surrounding tissues. **Methods:** In this study, we leverage reinforcement learning (RL) to enhance treatment planning with the aim of improving the adaptability and robustness of RL agents given the inherent inaccuracies in tumor growth models. A 2D simulation model of tumor growth is employed, where tabular RL techniques are used to determine the optimal treatment strategies. We emphasize the significance of tissue damage predictions and incorporate the Lyman NTCP model to assess treatment outcomes, analyzing complications across three simulated body sites: the rectum, head and neck and lung. **Results:** For all the tumor sites, the RL approach significantly reduces healthy tissue damage by 10.7%, 49.1% and 37.5%, respectively, for rectal, head and neck and lung cancers compared with the baseline treatment. **Conclusions:** The RL-based approach in radiotherapy not only achieves tumor eradication but also significantly reduces healthy tissue damage compared with traditional treatment methods. This study demonstrates the potential of reinforcement learning to optimize treatment planning in radiotherapy, offering a promising path towards more personalized and effective cancer treatments.

## 1. Introduction

Radiotherapy is a process where radiation doses are delivered in multiple sessions or fractions. This method seeks to effectively target and eradicate tumors while minimizing the dose to surrounding healthy tissues. Conventional fractionation schedules deliver the treatment in small doses (i.e., about 2 Gy) for several days (i.e., 30 fractions). However, the conventional fractionation paradigm is under scrutiny for its performance and outcomes. Recent studies hint towards a shift in this paradigm, advocating for more condensed and potentially effective treatment schedules (e.g., hypofractionation, with higher doses delivered in fewer fractions) [1,2,3], or yet more exotic combinations [4,5].

With the emergence of computational techniques and machine learning, groundbreaking strategies are on the horizon. Adaptive radiotherapy describes radiation plans adapted during treatment based on observed changes in the patient and promising enhanced treatment planning [6,7]. Furthermore, recent advances in mathematical modeling and in the understanding of tumor microenvironmental mechanisms now allow for the simulation of tumor growth and response to therapy, offering powerful tools to optimize treatment strategies. These models range from agent-based frameworks to complex multi-scale systems that incorporate tumor kinetics, metabolism, cellular signaling pathways and microenvironmental interactions [8,9,10,11].

Reinforcement learning (RL), a branch of machine learning, deploys an agent to interact with its environment in pursuit of optimal decision-making. In the context of radiotherapy, RL can be utilized to formulate treatment strategies, adapting dynamically to observed patient-specific conditions [12,13,14]. In a previous work led by Grégoire Moreau [4], the potential of RL was explored to optimize the total dose radiated on a patient, number of fractions, treatment time and successful tumor eradication. However, this exploration predominantly centered on optimizing the treatment’s efficiency and effectiveness while overlooking post-treatment complications.

The primary aim of this research is to harness tabular reinforcement learning (when the action–state function can be defined as a table) to determine the daily dosage in hypofractionated radiotherapy treatment planning. While RL has been explored in radiotherapy, the existing studies often lack integration with comprehensive models that assess normal tissue complications, such as the Normal Tissue Complication Probability (NTCP) model. This gap limits the clinical applicability of RL-based approach. Our study addresses this by integrating a 2D in silico tumor growth model with dose–volume histogram (DVH) analysis and the Lyman NTCP model [15]. This integration enables a comprehensive comparison between RL-based treatments and traditional counterparts, focusing on the total administered dose, number of fractions required for tumor eradication and the likelihood of complication occurrences for specific organ locations. This nuanced approach not only underscores the ways in which RL agents move towards hypofractionated treatments but also paints a comprehensive picture of their decision-making processes in balancing treatment optimization against potential complications.

The second aspect of our research focuses on evaluating the robustness and adaptability of our RL agents when confronted with errors in the tumor growth model. Our objective is to develop RL-driven solutions that maintain efficacy despite such errors. To achieve this, we systematically train and validate our agents across a broad spectrum of model parameters, ensuring that they can reliably navigate and respond to varying degrees of uncertainty within the model.

## 2. Related Works

### 2.1. Tumor Growth Model

Using mathematical models to predict tumor response to radiotherapy can significantly enhance the quality of treatment plans and even create specific treatments including patients’ characteristics [16,17,18,19,20].

O’Neil [21] proposed a detailed agent-based in silico model using lattice–gas cellular automata. This model can be defined by the tuple (*G*, *E*, *U*, *f*), where each element corresponds to a distinct aspect of the system:*G*: A grid of cells, shaping the model’s spatial structure.*E*: The various phases or cycles each cell might undergo, such as growth, division, or death.*U*: The neighboring cells, representing the local environment for interaction.*f*: A rule determining each cell’s behavior and state transition based on its current state and the states of its neighbors.

Expanding upon O’Neil’s foundational work, Jalalimanesh et al. [22] put forward an enhanced agent-based model by introducing oxygen as a primary nutrient. By simulating individual agents (e.g., cells) and their interactions, agent-based models provide valuable insights into system dynamics. This leads to a model, used by Moreau et al. [4], that recognizes both glucose and oxygen as essential nutrients. This approach underscores the significance of both elements in aerobic respiration, a process that is integral to cell function. In the subsequent sections of this article, we will use the term “tumor growth model” to refer to our “in silico agent-based model” in order to avoid redundancy regarding the in silico term and prevent any confusion with the term “agent” as used in the context of RL.

Alongside these cellular modeling advancements, the linear–quadratic (LQ) model emerged as a pivotal predictor of ionizing radiation’s effects on cell survival. This mathematical framework calculates the cell survival fraction (SF) post-exposure to different radiation doses, with α and β being the linear and quadratic components of radiation-induced cell damage, respectively, and *D* being the dose received by the cell:(1)$$\begin{array}{c} SF\left(D\right)=exp\left(-\alpha D-\beta D^{2}\right). \end{array}$$

Jalalimanesh et al. [22] further refined the LQ model by introducing radiosensitivity $\gamma_{r}$ and an Oxygen-Modifying Factor (OMF). This definition is used in this work:(2)$$\begin{array}{c} SF\left(D\right)=exp[\gamma_{r}\left(-\alpha \;\mathrm{OMF}\;D-\beta {\left(\mathrm{OMF}\;D\right)}^{2}\right)]. \end{array}$$

Emami et al. [23] undertook a comprehensive literature review, amassing data on normal tissue radiation tolerance. This work laid the foundation for TS Kehwar’s determination of the LQ model values for various cancer locations [24].

### 2.2. Lyman–Kutcher–Burman Model

The Lyman–Kutcher–Burman (LKB) model [15] predicts the likelihood of radiation-induced complications in normal tissues after a fractionated radiotherapy treatment. Central to the LKB model is the equivalent uniform dose (EUD) concept, which equates a non-uniform dose distribution to an equivalent uniform dose with a similar biological effect. Dennstadt et al. [25] presented a systematic review concentrating on the LKB model’s parameters.

## 3. Materials

### 3.1. In Silico Tumor Growth Model

The environment formulation is the same as in the previous work conducted by Moreau et al. [4]. The model representation is composed of four 50 × 50 2D grids, as shown in Figure 1, where each pixel on the grid corresponds to

The types of cells within this pixel. Healthy and cancer cells can co-exist on the same pixel. In such cases, the pixel is displayed in red to denote the presence of cancer cells.The cell density on the grid.The glucose concentration.The oxygen concentration.

The following cell cycle is used in the tumor growth model:Gap 1 (G1): Cell growth and preparation for DNA replication (11 h).Synthesis (S): DNA replication (8 h).Gap 2 (G2): Cell growth and preparation for mitosis (4 h).Mitosis (M): Formation of two daughter cells (1 h).

**Figure 1 biomedicines-13-01367-f001:**
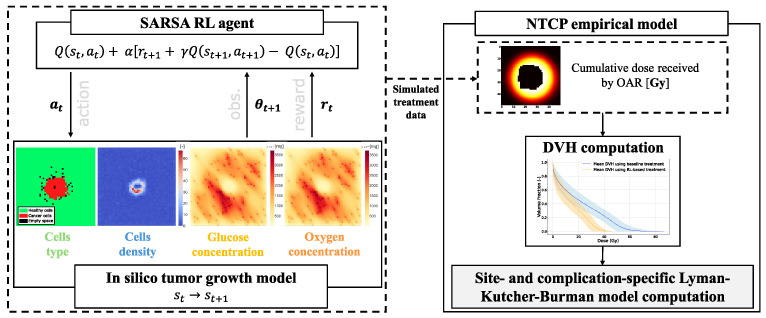
Overall workflow diagram for each treatment simulation. At each timestep *t*, an action, denoted $a_{t}$, results in a transition from state $s_{t}$ to state $s_{t+1}$. Subsequently, the environment yields a reward $r_{t}$ and an observation $\theta_{t+1}$. The data obtained are used to compute the OARs DVHs and, thanks to the LKB model, the resulting NTCP.

In the initial phase of the simulation, 1000 healthy cells are randomly distributed on the grid, and a single cancer cell is placed at the center. As the simulation progresses, the cancer cell undergoes replication, causing the tumor to grow incrementally. Once the number of cancer cells reaches a threshold of 9000, the radiotherapy treatment is initiated. This threshold was determined empirically to balance computational efficiency with the presence of a sufficiently large tumor. Additionally, each cell is given its individual metabolic efficiency parameters at initialization. These parameters are drawn from normal distributions that reflect biological variability and differ depending on the cell type (healthy or cancerous) and the metabolic process considered as follows:An oxygen consumption efficiency, which reflects the cell’s oxygen usage during DNA replication. For both healthy and cancer cells, oxygen consumption is modeled as a normally distributed random variable, $N(2.16\times 10^{-9},0.72\times 10^{-9})$ [mg/cell/hour]. These values are based on the measurements reported by O’Neil [21].A glucose absorption efficiency, representing the cell’s glucose uptake rate, which is vital for DNA replication. This parameter is sampled from-$N(3.6\times 10^{-9},1.2\times 10^{-9})$ [mg/cell/hour] for healthy cells.-$N(5.4\times 10^{-9},1.8\times 10^{-9})$ [mg/cell/hour] for cancer cells.

The cellular environment comprises three different endpoint states:Treatment is successful if all cancer cells die due to lack of nutrients or radiation. Even one remaining cancer cell can lead to a new tumor.Treatment fails if radiation therapy results in a growing tumor. If the number of healthy cells falls below 10, the simulation ends, indicating failure.Treatment duration is limited to 1200 h (50 days). Exceeding this time limit results in the termination of the simulation.

The LQ model is used to reflect the effects of radiation on cell survival. We use a stochastic approach in which a random number between 0 and 1, following a uniform distribution, is generated. If this number exceeds the SF calculated by the LQ model, the cell is considered dead. Our study is entirely based on an in silico framework and does not rely on any external or clinical dataset. Instead, the tumor microenvironment and response to treatment are simulated in real time through an in silico agent-based model, whose parameters are derived from previously published literature [24,25,26]. The reinforcement learning agent interacts directly with this simulated environment, and transitions are computed on-the-fly during training without storing an external dataset. Table 1 summarizes the parameters of the tumor growth model.

### 3.2. LQ Model Parameters

This section outlines the methodology used for selecting the values of α and β (parameters of the LQ model) for both normal and tumor tissues. The term “normal tissues” is used to refer to the tissues of the organ at risk (OAR). Since the values of α and β are specific to the tumor site, we selected three organs to study: rectum, head and neck and lung. Table 2 summarizes the α and β values for the different organs selected.

The values of $\alpha_{norm}$ and $\beta_{norm}$ for the LQ model are derived from the work of TS Kehwar [24], which in turn sourced these values from the research of Enami et al. [3]. This methodology ensures the appropriate parameterization of the LQ model, aiding in the accurate representation of radiation response across different tissue types.

For tumor tissues, our approach is based on the clinical α/β ratio of approximately 10, as reported in van Leeuwen et al. [26] for early-responding tissues. Rather than selecting fixed values, we identified pairs of (α,β) that respect this ratio and lead to complete tumor eradication under conventional radiotherapy protocols. We explored a range of prescribed total doses and fitted α values accordingly. The associated β values were then derived to preserve the targeted α/β ratio. This methodology ensures consistent values with clinically accepted dose prescriptions for the targeted organ in conventional therapy.

### 3.3. Reinforcement Learning

Reinforcement learning is rooted in the Markov Decision Process (MDP), a mathematical framework characterized by Sutton and Barto [27]. The framework represented in Figure 1 provides a formal structure for environments where an agent makes decisions. It is especially useful for sequential decision-making scenarios. MDPs comprise the following:States $s_{t}\in S$, with *S* representing the state space.Actions $a_{t}\in A$, with *A* as the action space.Rewards $r_{t}$ constrained within [0, $R_{max}$], where $R_{max}$ is normalized to 1.A transition function $T\left(s,a,s^{\prime }\right):S\times A⟶\left[0,1\right]$ denoting the probability of reaching state *s’* from state *s* after action *a*.A discount factor γ, ranging between [0,1).

The SARSA algorithm offers a means to learn a policy aiming to maximize the sum of expected future rewards. Rather than immediately approximating the optimal policy, SARSA learns the Q-value function, denoted as $Q(s_{t},a_{t})$, incrementally. This function estimates the expected cumulative future reward for each state–action pair given that the agent adheres to a certain policy. The Q-value function is updated iteratively as the agent explores the state–action space following the equation(3)$$\begin{array}{c} Q\left(s_{t},a_{t}\right)\leftarrow Q\left(s_{t},a_{t}\right)+\eta \left[r_{t+1}+\gamma Q\left(s_{t+1},a_{t+1}\right)-Q\left(s_{t},a_{t}\right)\right]. \end{array}$$

$Q(s_{t},a_{t})$ represents the current estimate of the Q-value for the current state–action pair.$r_{t+1}$ is the immediate reward after taking action $a_{t}$ in state $s_{t}$.$Q(s_{t+1},a_{t+1})$ is the estimated Q-value for the next state–action pair.η is the learning rate, which dictates the step size in updating the Q-value.γ is the discount factor, which discounts future rewards.

The SARSA algorithm uses the Q-value of the action actually taken (on-policy) during the next timestep. Specifically, it learns an action–value function $q_{\pi }\left(s,a\right)$ for the current policy π.

### 3.4. NTCP Empirical Model

The following steps are performed to calculate the NTCP:The process starts by obtaining the DVH data for the OAR. The DVH represents the distribution of dose within the tissue volume. It takes the form of a histogram representing the volume fraction (ordinate) irradiated by a given dose (abscissa).The generalized equivalent uniform dose (gEUD) is calculated from the DVH data. The gEUD is a single dose value that accounts for the variable dose distribution across the volume of a tissue or organ. In other words, it is the dose that, if given uniformly, would lead to the same radiobiological effect.Once the gEUD is calculated, it is plugged into the LKB model to obtain the NTCP.

Based on the DVH data, the gEUD can be computed as follows:(4)$$\begin{array}{c} gEUD={\left(\sum_{i}V_{i}\;D_{i}^{1/n}\right)}^{n} \end{array}$$
where $V_{i}$ is a fraction indicating the proportion of the complete irradiated volume that has received at least the cumulative dose (over the treatment) $D_{i}$. The parameter *n* is a volume-effect parameter that describes the tissue’s sensitivity to the dose distribution. The sum is performed over all the bins “*i*” composing the DVH. Then, the NTCP value is computed as follows:(5)$$\begin{array}{c} NTCP=\frac{1}{\sqrt{2\pi }}\;\int_{-\infty }^{\tau }\exp\left(\frac{-x^{2}}{2}\right)\;dx. \end{array}$$
where $\tau =\frac{gEUD-TD_{50}}{m\;TD_{50}}$. $TD_{50}$ is the uniform dose that results in a 50% complication probability, and *m* is a parameter that describes the slope of the dose–response curve. The parameters $TD_{50}$, *m* and *n* are specific to the cancer locations and complications under study. However, it is important to note that the precise definition of the endpoint can vary significantly across different studies. The chosen values for the different cancer locations are summarized in Table 3. The parameters *m* and $TD_{50}$ are mean values based on the work of F Dennstädt et al. [25], who conducted a comprehensive meta-analysis, while *n* is intentionally chosen a much smaller value than the literature to show a lower bound for the NTCP.

### 3.5. Dose–Volume Histogram

The calculation of NTCP relies primarily on the computation of DVH data for the OAR. To maintain accurate tracking throughout the process, a cumulative dose is recorded for each voxel during the course of the simulation. Since DVHs are typically expressed in terms of relative volumes, two distinct masks are generated to distinguish the different regions on the grid. One represents the complete volume of the OAR, while the other represents the Target Volume (TV). The two masks are computed before the beginning of the radiotherapy treatment simulation and are called the OAR and TV masks. Each pixel containing a cancer cell constitutes, together, a 2D mask of the grid where the tumor is located. Then, this mask is dilated by two pixels to obtain the TV mask. The OAR mask is computed as the logical “NOT” of the TV mask.

## 4. Methods

The primary goal of this research is to evaluate the efficacy of RL-based fractionation schedules for radiotherapy treatments. This involves two main phases: training the RL agent and subsequently evaluating its performance across various cellular environments. This section explains the methodology employed to choose the parameters of the tumor growth model, which plays a crucial role in the training and testing of our agents. The codes used to obtain our results are available at https://github.com/meghislain/RL_fractionation (accessed on 29 May 2025).

### 4.1. Integrating Reinforcement Learning with the Tumor Growth Model

Our tumor growth model provides an environment where an RL agent can be trained. The agent aims to formulate an optimal treatment strategy that ensures the elimination of cancer cells while conserving healthy cells. While the in silico tumor growth model evolves with a timestep denoted *t* on an hourly basis, the RL agents are allowed to observe and take actions only every 24 h (=24 timesteps). The state space of the MDP is constructed from the grid of the tumor growth model. Each state encapsulates the number of healthy and cancer cells present on the grid. The agent can administer, every 24 h of simulation, a radiation dose *d* of 1, 2, 3, or 4 Gy representing the action space. The reward function (R${\,}_{k}$), depending on the timestep *k*, as outlined in (Equation 6), has been designed to motivate the agent towards the dual goal of preserving healthy cells and annihilating cancer cells while minimizing the dose administered. Specifically, it was reutilized from the work of Moreau et al. [4]:(6)$$\begin{array}{c} r_{k}=-\frac{d_{k}}{400}+\frac{c_{k}-5h_{k}}{100,000} \end{array}$$
where $d_{k}$ is the radiation dose, $c_{k}$ is the number of killed cancer cells, and $h_{k}$ is the number of killed healthy cells. Each variable is evaluated at timestep *k*, a multiple of 24 h.

### 4.2. Radiotherapy Treatment Assessment and Key Performance Indicators

The assessment of radiotherapy treatment effectiveness is based on five Key Performance Indicators (KPIs), which provide insights into both the treatment’s success and its associated risks:Success Rate (SR) [%]: SR quantifies the treatment’s overall success, calculated as the percentage of simulations that resulted in complete tumor eradication relative to the total number of simulations conducted.NTCP [%]: NTCP represents the estimated likelihood of radiation-induced complications in healthy tissue, expressed as a percentage. This metric helps to predict potential adverse effects in specific organs.Dose [Gy]: This refers to the mean total radiation dose administered across all simulations, providing an average measure of the radiation intensity delivered in Grays (Gy).Fractions [-]: The mean number of fractions (treatment sessions) per simulation, offering insight into the fractionation strategy used during treatment.Duration [h]: The mean treatment duration across all simulations, expressed in hours, providing an assessment of the overall length of the treatment regimen.

KPIs are averaged across 100 simulations conducted with identical parameters and conditions to account for model variability. As described in Section 3, randomness is inherent in the models used in this work to ensure the robustness of our RL agents. We then provide the results based on the average obtained across the simulations. DVHs are also provided, showcasing the mean values and their standard deviations across the different treatment methods.

For RL-based treatments, we additionally analyze the prescribed radiation doses over the course of the treatment. We present the average dose ($\mu_{\mathrm{dose}}$) and its standard deviation ($\sigma_{\mathrm{dose}}$), computed across all simulations for each time the RL agent administers a dose. Finally, we pinpoint the 99% Confidence Interval (CI) for NTCP values, corresponding to the interval $\left[\mu_{\mathrm{NTCP}}-3\;\sigma_{\mathrm{NTCP}},\mu_{\mathrm{NTCP}}+3\;\sigma_{\mathrm{NTCP}}\right]$.

### 4.3. Robustness Against Cellular Model Errors

To evaluate the robustness of our approach, we investigate the impact of potential errors in the cellular model. Specifically, we focus on two key aspects:Nutrient consumption rate of cancer cells: We examine how changes in the nutrient consumption rates of cancer cells affect treatment outcomes, assessing whether these variations lead to significant deviations in the results.LQ model parameters: We investigate the impact of variations in the $\alpha_{\mathrm{tumor}}$ and $\beta_{\mathrm{tumor}}$ parameters on the model’s performance. These parameters, which are crucial for determining the effectiveness of the radiation dose, are selected based on the ranges derived from the literature review. These ranges correspond to-Rectum: $\alpha_{tumor}\in \left[0.065-0.465\right]$ and $\beta_{tumor}\in \left[0.005-0.103\right]$.-Head and neck: $\alpha_{tumor}\in \left[0.25-0.33\right]$ and $\beta_{tumor}\in \left[0.025-0.033\right]$.-Lung: $\alpha_{tumor}\in \left[0.25-0.30\right]$ and $\beta_{tumor}\in \left[0.025-0.030\right]$.

An agent is considered robust when it achieves an SR of 100% regardless of the variations in the parameters under study.

## 5. Results

### 5.1. RL-Based Treatments for Different Cancer Locations

The performance of the RL-based treatments compared with the baseline across all the cancer locations (rectum, head and neck and lung) is summarized in Table 4, which outlines the KPIs.

For lung cancer, the NTCP is notably elevated due to the choice of $TD_{50}=29.88\;$ Gy, as reported in Table 4. Since the treatment frequently exceeds this threshold, the NTCP increases. The behavior of the RL-based treatments in relation to dose fractionation is illustrated in Figure 2. This shows how the RL agent adjusts radiation doses dynamically based on the cancer’s response for rectal, head and neck and lung cancers, respectively, in Figure 2a,c,e. The counts of both healthy and cancer cells are emphasized to provide clearer insight into the agent’s choices. Figure 2b,d,f present the dose–volume histograms (DVHs) of organs at risk (OARs), comparing the baseline treatment with the RL-based approach for rectal, head and neck and lung cancers, respectively.

### 5.2. Generalization of Trained Agents or Robustness of Trained Agents

To assess the robustness of our RL-trained agents across varying tumor-specific parameters, we tested the models against different values of $\alpha_{tumor}$ and $\beta_{tumor}$. The results and corresponding KPIs are given in Table 5. When $\alpha_{tumor}=0.065$ and $\beta_{tumor}=0.005$, the success rate (SR) dropped to 0%, while the NTCP rose to 32%, indicating that the agent could not meet the pre-defined objectives under these conditions. This drop in performance can be attributed to the increased radiation resistance of cancer cells due to the altered values of $\alpha_{tumor}$ and $\beta_{tumor}$. For values in the same range as the ones used in training, the desired objective is reached (SR of 100%) with minimal NTCP values and a small number of fractions. Recognizing the challenge of finding precise values from the literature for the LQ model in use for head and neck and lung cancers, we chose to keep the ratio $\alpha_{tumor}/\beta_{tumor}$ constant at 10. Consequently, we varied the values for $\alpha_{tumor}$.

### 5.3. Retraining

Our primary goal is to deliver treatments optimized for tumors of various degrees of invasiveness. To achieve this, we focused on refining our agent by retraining it in tailored environments, characterized by the new pairs of $\alpha_{tumor}$ and $\beta_{tumor}$ used in Table 5. The outcomes of this targeted retraining are shown in Table 6.

For all the cancer sites, the agent specification did not improve their performance beyond what is depicted in Table 5. The performance was either the same or slightly but not significantly better (the NTCP was equal to or just lower than the results without retraining).

## 6. Discussion

The results appear to indicate that the RL-based treatment has the potential to offer advantages over the baseline in terms of safety (NTCP) and efficiency (dose, fractions and duration). This suggests that exploring the application of RL within radiation therapy might yield benefits in enhancing patient safety and optimizing the treatment protocols, although these findings need to be interpreted with caution given the assumptions made in our study. The DVHs of the OARs for the various cancer locations seem to lean in favor of the RL-based treatments over the baseline outcomes (Figure 2b,d,f). In our observations, the DVH curves for the RL-based treatments often lie below those of the baseline treatments.

### 6.1. General Agent Behavior

In the preliminary stages before radiotherapy is initiated, an average of 4200 healthy cells juxtaposed with a significantly larger number of 9500 cancer cells tends to be the standard observation. A consistent strategy adopted by our agent, irrespective of the cancer location and potential complications, is to deliver an initial radiation dose of 4 Gy (Figure 2a,c,e). On average, only 12% of the healthy cells are killed after the first radiation dose versus 82% of the cancer cells. The potent efficacy of this first radiation dose can be attributed to two primary factors. Firstly, this is due mainly to the high concentration of cancer cells and the accuracy of our radiation doses, which are always precisely targeted on the tumor. Secondly, this behavior is also due to our reward function, which, even if it promotes low dose values, also mainly rewards the effectiveness of killing cancer cells.

On analyzing the radiotherapy patterns across the three distinct cancer locations, a discernible trend emerges. The first half of the treatment trajectory predominantly features high radiation doses (consistently exceeding 3 Gy). In contrast, the second half of the treatment is characterized by a concerted effort to mitigate the dose intensity. A notable observation is that the radiation doses delivered during the second and third sessions are invariably lower than that of the fourth session. While these doses still fall within the higher range, the agent is clearly modulating the radiation strategically to afford healthy cells a recovery window, especially in the aftermath of the potent 4 Gy dose administered initially. This trend is validated by our plots (Figure 2a,c,e), which consistently demonstrate an uptick in healthy cell counts following the fourth radiation dose compared with the two preceding sessions.

Once the agent has killed the majority of the cancer cells while maintaining a relatively high number of healthy cells, it starts to decrease the radiation doses slightly over the treatment time in order to limit the side effects on the organ at risk. The agent exhibits a more conservative approach, leveraging moderate radiation doses (ranging between 2 and 3 Gy) with two objectives in mind: methodically eliminating the residual tumor cells and facilitating an environment conducive to the proliferation of healthy cells.

Toward the end of the treatment, there are instances where the agent, in an attempt to ensure a conclusive treatment, reverts to higher radiation doses. This strategic choice underscores the agent’s preference for a more immediate and potent dose rather than prolonging the treatment. Prolongation could result in augmented irradiation exposure for the OARs, which is detrimental in the long run. Additionally, Figure 2 depicts a significant standard deviation of the radiation dose decided by the agent at the end of the RT treatment that can be explained by the agent’s objective to eradicate the tumor while giving the patient a minimum radiation dose. The inherent variability in the tumor growth model leads to a great variety of scenarios, causing the agent to face a broad spectrum of remaining cancer cells at treatment end. The agent’s decisions are “simulation-specific” towards the end of an RT treatment.

### 6.2. Overall Results

The results, as presented in Table 4, highlight the superior performance of the RL-based approach in treating rectal cancer patients. The RL agent achieves a 100% SR while significantly minimizing the NTCP to nearly 0%. This stands in stark contrast to the 10.74% noted with the traditional baseline treatment. The RL-guided treatment thus reduced the NTCP by 10.74% compared with the traditional baseline treatment. Additional KPIs reinforce the outstanding efficacy of the RL-based treatment, indicating considerable improvements in every metric. The same observations can be made for the two other cancer locations (Table 4), with a decrease in the NTCP from 80.42% to 31.29% (−49.1%) and from 99.44% to 61.9% (−37.5%) for head and neck and lung cancers, respectively. Noting that the selected NTCP model used values of n<1, which assign greater weight to high-dose subvolumes and reflect low tolerance to local hotspots, the observed reductions are especially encouraging. A value of n=1 would treat all the parts of the volume equally, making our results even more notable given the stricter dose-sensitivity assumptions. The total treatment time is also reduced by a factor of three, on average, with the RL-based treatment fractionation.

However, ensuring the robust adaptability of our agent across the full spectrum of values for $\alpha_{tumor}$ and $\beta_{tumor}$ presents a challenge. Utilizing extremely low values for $\alpha_{tumor}$ and $\beta_{tumor}$, as shown in Table 5, is particularly problematic as it results in exceptionally high survival fractions of cancer cells, even when high irradiation doses are applied. The two main reasons for this lack of robustness facing extremely low $\alpha_{tumor}$ and $\beta_{tumor}$ values are (1) the constraints on the agent’s action, where it is restricted to using radiation doses up to a maximum of 4 Gy, and, (2) the fact that the number of cancer cells killed between two consecutive radiation doses is smaller than the number of cells reproduced, leading to exponential growth of the cancer site. Other cancer locations (head and neck and lung cancers) are not subject to such extremely low $\alpha_{tumor}$ and $\beta_{tumor}$ values as rectal cancers, leading to the complete robustness of our agents, which always achieve SR values of 100% (Table 5). Therefore, even when it is trained on a specific cancer site, our RL agent exhibits strong performance in similar environments and is therefore robust to moderate alpha–beta parameter errors.

We retrain the agents for each cancer site with site-specific alpha–beta parameters to assess potential improvements in the KPIs. However, this did not result in more optimal behavior: Table 6 does not reveal significant differences between the retrained agent and the initial one. This can be attributed to the high performances proposed by the initial agent, making improvement difficult. Indeed, conventional treatments consist of twenty-eight or thirty fractions to complete tumor eradication. Our agents propose RL-based treatments delivering between nine and sixteen fractions, a number that is difficult to reduce even by a retrained agent on the specific environmental conditions. The failure of the retrained agent to completely eradicate rectal cancer tumors with very low alpha levels can be attributed to the fact that these alpha values fall outside the effective treatment range, as discussed earlier.

### 6.3. Nutrient Consumption Variations

As previously mentioned, we tested the robustness of our agents with respect to variations in the cancer cells’ nutrient consumption rates. Our observations showed no significant shifts in the KPIs, leading us to conclude that these variations do not influence the treatments administered by our agents.

### 6.4. Limitations

The variations in the Lyman model parameter values can primarily be attributed to differences in the classification systems used and the specific definitions of the endpoints within those systems. These differences may arise from variations in the methodologies, criteria, or standards applied, leading to discrepancies in the outcomes or interpretations of the model. The exact definition of the Lyman model itself influences the values of the clinically derived parameters. Some studies focus on fitting the model for specific needs, leading to large ranges of collected values.

The values of α and β respect the idea of a ratio of $\frac{\alpha }{\beta }≅10$ for early-responding tissues, such as tumorous tissues, and a ratio of $\frac{\alpha }{\beta }≅3$ for late-responding tissues, such as normal tissues. Nevertheless, when delving into the precise values of $\alpha_{tumor}$ and $\beta_{tumor}$ corresponding to cancerous tissues, the literature presents a diverse spectrum of values. This observed variability can be attributed to two primary factors: the heterogeneity inherent in tumor types and the methodologies employed to ascertain these parameters. Different tumor classifications can manifest distinct radiobiological responses, and the techniques used to measure these responses may also introduce variations.

One important limitation of our approach is the use of a 2D in silico tumor model. While this allows for computational efficiency and clarity in spatial dynamics, it does not capture the full complexity of a real 3D tumor structure. In particular, the volumetric tumor geometry, dose absorption and 3D diffusion processes are simplified.

### 6.5. Future Work

Delving deeper into the realm of hypofractionation and considering higher radiation doses could offer promising avenues for advancement. By incorporating a broader spectrum of doses, we stand a chance to understand better and harness the potential therapeutic benefits and outcomes. This is particularly salient when considering the reward function in our model. If we were to link the reward function with the NTCP explicitly, the algorithm would be provided with a more direct feedback mechanism than the more indirect method of monitoring the death of healthy cells.

Furthermore, the adoption of more realistic cellular models is imperative for the evolution and enhancement of RL algorithms in this context. Our current models, while effective to some extent, offer solutions with relative ease. A more intricate and accurate cellular model might pose challenges initially but would also push the RL algorithms to uncover novel strategies and solutions that could be closer to real-world applications. The parameters of such models could be optimized to generate a more faithful representation of the tumoral tissue on the basis of histology or even in vivo histology obtained by diffusion-weighted magnetic resonance imaging [28]. Moreover, in our current model, the RL agent operates within a simulated environment where biological information is represented numerically for each pixel. However, in clinical settings, such precise cellular counts or oxygen concentrations are not directly observable. To transition this model towards clinical applicability, it is imperative to integrate data from clinically accessible modalities. The relevant methods may include functional imaging, such as positron emission tomography (i.e., FDG-PET) for glucose metabolism, hypoxia-specific tracers for oxygenation, cone-beam computed tomography or magnetic resonance imaging for tumor volume and magnetic resonance spectroscopy or biopsy for cell type identification. Blood samples can also yield insights into circulating cells and other biomarkers. Incorporating information available in medical data into the RL framework would enable the agent to make decisions based on real-world observations but necessitate careful consideration of logistical aspects.

In particular, transitioning from a 2D to a fully 3D model would offer a more biologically realistic environment for tumor simulation and could significantly impact both oxygen/glucose distribution and treatment response modeling while being optimally tailored for actual clinical scenarios. Although the model does not explicitly simulate intracellular signaling pathways, it incorporates the effect of angiogenesis in a simplified form through stochastic spatial variability in nutrient availability. Future extensions could involve coupling such macroscopic behavior with pathway-level models. For instance, instead of simulating the effects of conventional radiotherapy alone, we could integrate the intervention of additional treatment modalities. Notably, studies have explored the synergistic effects of afatinib, an EGFR inhibitor, on the radiosensitivity in non-small-cell lung cancer patients [29]. Similarly, emerging approaches like oxygen-independent radiodynamic therapy represent innovative strategies that combine radiotherapy with immunomodulatory effects [30]. The integration of those treatment combinations could open new avenues for innovative therapeutic strategy modeling in oncology.

## 7. Conclusions

This study has highlighted the significant potential of RL in advancing the field of radiotherapy. By focusing on the division of radiation doses into multiple sessions, we aimed to optimize treatment planning strategies that both improve efficacy and reduce complications. A 2D simulation model of tumor growth served as a foundation for employing tabular RL techniques to determine the optimal treatment strategies. A notable aspect of our approach was the incorporation of the Lyman NTCP model, a well-established tissue complication model, to assess the treatment outcomes. This integration was crucial in enhancing the adaptability and robustness of our RL agents, particularly given the inherent challenges in eradicating the tumor effectively while preserving the surrounding healthy tissues.

Our findings are promising, showing the efficacy of the RL approach for three simulated body sites: the rectum, head and neck and lung. The RL-based treatment achieves complete tumor eradication and significantly minimizes healthy tissue damage. The results show reductions in the NTCP of 10.74%, 49.1% and 37.5% for rectal, head and neck and lung cancers, respectively, compared with the traditional baseline treatment (Table 4). Additionally, the RL-based approach reduced the overall treatment duration. We also observed a consistent dose delivery pattern across scenarios, with larger doses being delivered during the initial treatment fractions followed by a gradual decrease. This trend aligns with the emerging concepts in adaptive hypofractionation, where initial aggressive dosing is used to target rapidly dividing cells, followed by more conservative fractions to minimize collateral damage.

Looking ahead, the potential for RL in radiotherapy is vast. Transitioning from simplified 2D environments to biologically realistic 3D models could substantially improve the fidelity of in silico results. Furthermore, enabling agents to operate based on real clinical inputs, such as segmented medical images, rather than abstract cell counts would bridge the gap between simulation and practice. The next steps involve validating these findings through clinical trials, expanding to more diverse tumor types and patient populations and exploring the integration of RL with multimodal cancer therapies. These advances could revolutionize oncology treatment planning and enable more personalized and effective cancer therapies.

## Figures and Tables

**Figure 2 biomedicines-13-01367-f002:**
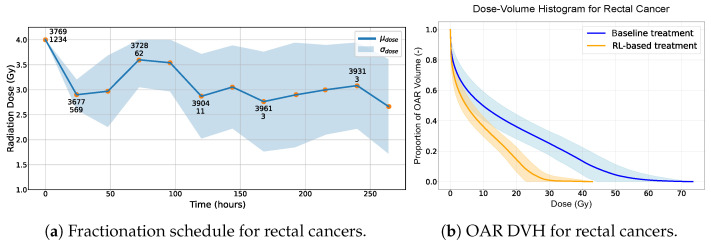
(**a**,**c**,**e**) Radiation dose over time provided by the RL-based treatment. The standard deviation $\sigma_{dose}$ becomes less indicative of the overall treatment variation when 216 h is exceeded since only 12% of the simulations reach this threshold. Two numbers are present for some data points. The first indicates the average number of healthy cells, and the second represents the average number of cancer cells, observed after administering a specific radiation dose across all simulations. (**b**,**d**,**f**) Dose–volume histogram of the OAR using RL-based and baseline treatments. The standard deviation becomes less indicative of the overall treatment variation when a total dose of 60 Gy is exceeded as not all simulations reach this threshold.

**Table 1 biomedicines-13-01367-t001:** Tumor growth model parameters.

Parameters	Theoretical Values	Units
Starting number of healthy cells	1000	-
Starting number of cancer cells	1	-
Starting number of nutrient sources	100	-
Starting glucose level	$1\times 10^{-6}$	mg
Starting oxygen level	$1\times 10^{-6}$	mL
Average glucose absorption (healthy)	$3.6\times 10^{-8}$	mg/cell/h
Average glucose absorption (cancer)	$5.4\times 10^{-8}$	mg/cell/h
Average oxygen consumption (healthy)	$2.16\times 10^{-8}$	mL/cell/h
Average oxygen consumption (cancer)	$2.16\times 10^{-8}$	mL/cell/h
Critical oxygen level	$3.88\times 10^{-8}$	mL/cell
Critical glucose level	$6.48\times 10^{-8}$	mg/cell
Quiescent oxygen level	$10.37\times 10^{-8}$	mL/cell
Quiescent glucose level	$1.728\times 10^{-8}$	mg/cell

**Table 2 biomedicines-13-01367-t002:** Parameter values of the linear–quadratic model for the different organs selected.

Cancer Location	$\alpha_{\mathrm{tumor}}$	$\beta_{\mathrm{tumor}}$	$\alpha_{\mathrm{norm}}$	$\beta_{\mathrm{norm}}$
	**(Gy${\,}^{-1}$)**	**(Gy${\,}^{-2}$)**	**(Gy${\,}^{-1}$)**	**(Gy${\,}^{-2}$)**
Rectum	0.315	0.0662	0.0484	0.0124
Head and neck	0.330	0.029	0.0341	0.0114
Lung	0.325	0.0325	0.0637	0.0168

**Table 3 biomedicines-13-01367-t003:** Chosen parameters for Lyman’s NTCP empirical model [25] for the different organs and complications under study.

Cancer Location	Endpoint	${\mathrm{TD}}_{50}$	*m*	*n*
Rectum	Rectal bleeding	80.10	0.150	0.15
Head and neck	Xerostomia	40.28	0.408	0.01
Lung	Symptomatic pneumonitis	29.88	0.400	0.15

**Table 4 biomedicines-13-01367-t004:** Key performance indicators obtained with the baseline and RL-based treatments for all cancer locations.

		SR	NTCP	NTCP CI	Dose	Fraction	Duration
	**Method**	**(%)**	**(%)**	**(%)**	**(Gy)**	**(-)**	**(h)**
Rectum	Baseline	100.0	10.74	[7.55,14.1]	50.99	28.33	679.92
	RL-based	100.0	0.006	[0.00,0.02]	25.56	7.94	190.56
Head and neck	Baseline	99.0	80.42	[78.3,82.7]	59.42	29.71	713.04
	RL-based	100.0	31.29	[27.1,31.2]	34.04	10.14	243.36
Lung	Baseline	100.0	99.44	[99.2,99.6]	57.02	28.51	684.24
	RL-based	100.0	61.90	[58.3,65.0]	35.01	11.17	268.08

**Table 5 biomedicines-13-01367-t005:** Key performance indicators obtained with RL-based treatment for all cancer locations tested with various values of $\alpha_{tumor}$ and $\beta_{tumor}$.

	$\alpha_{\mathrm{tumor}}$	$\beta_{\mathrm{tumor}}$	SR	NTCP	Dose	Fraction	Duration
	(${\mathrm{Gy}}^{-1}$)	(${\mathrm{Gy}}^{-2}$)	**(%)**	**(%)**	**(Gy)**	**(-)**	**(h)**
Rectum	0.065	0.005	0.0	32.02	63.06	42.46	1019.04
	0.265	0.054	100.0	0.003	32.12	9.84	236.16
	0.465	0.103	100.0	$4.91\times 10^{-5}$	16.19	4.86	116.64
Head and neck	0.25	0.025	100.0	62.63	48.95	14.71	353.04
	0.28	0.028	100.0	43.93	39.78	11.87	284.88
	0.33	0.033	100.0	30.96	33.73	10.06	241.44
Lung	0.25	0.025	100.0	97.65	50.54	15.77	378.48
	0.28	0.028	100.0	91.04	42.76	13.37	320.88
	0.30	0.030	100.0	82.12	38.03	11.91	285.84

**Table 6 biomedicines-13-01367-t006:** Key performance indicators obtained with a retrained agent for all cancer locations using various values of $\alpha_{tumor}$ and $\beta_{tumor}$.

	$\alpha_{\mathrm{tumor}}$	$\beta_{\mathrm{tumor}}$	SR	NTCP	Dose	Fraction	Duration
	**(Gy${\,}^{-1}$)**	**(Gy${\,}^{-2}$)**	**(%)**	**(%)**	**(Gy)**	**(-)**	**(h)**
Rectum	0.065	0.005	0.0	58.07	74.54	44.53	1068.7
	0.265	0.054	100.0	0.047	29.96	8.78	210.72
	0.465	0.103	100.0	$2.021\times 10^{-5}$	16.45	5.56	133.44
Head and neck	0.25	0.025	100.0	66.56	50.81	15.93	382.32
	0.28	0.028	100.0	46.71	41.14	12.47	299.28
	0.33	0.033	100.0	27.91	31.93	9.64	231.36
Lung	0.25	0.025	100.0	96.05	49.24	15.18	364.32
	0.28	0.028	100.0	86.83	40.85	12.75	306.00
	0.30	0.030	100.0	80.32	36.83	11.61	278.64

## Data Availability

The codes used to obtain our results are available at https://github.com/meghislain/RL_fractionation, accessed on 29 May 2025.

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
