# Peer review of "Optimal Fractionation Scheduling for Radiotherapy Treatments with Reinforcement Learning, Tumor Growth Modeling and Outcome Modeling"

_biomedicines, 2025, doi:10.3390/biomedicines13061367_

Round 1

Reviewer 1 Report

Comments and Suggestions for Authors

The manuscript entitled ‘Optimal fractionation scheduling for radiotherapy treatments with reinforcement learning, tumor growth modeling and outcome modeling’ intends to explore machine learning and tumor simulation models to demonstrate better efficacy of radiotherapy in terms of optimal dosage and radiotherapy treatment timing as suggested based on clinical data. The manuscript has significant potential in terms of the concept and relevance of the topic. However, there seems to be significant research gap in application of machine learning and understanding complicated microenvironment conditions such as metastatic cancers tissues. Yet, this is a significant advent with a hopeful insight in near future. The challenge yet remains to successfully demonstrate on in vivo models.

  1. The title of the subsection 2.2. is NTCP empirical model, however, the content of the subsection is of LKB model. Kindly crosscheck the content/subsection title.
  2. The title of the subsection 3.1. is Tumor growth model. However, it is essential to mention if it is in vivo, in vitro, or in silico model.
  3. Significant grammatical modification is needed in line 98-100
  4. What does two numbers (3.6 × 10−9, 1.2 × 10−9) indicate in line 124 and several others?
  5. Everywhere else, kindly mention the type of tumor growth model.
  6. What is the basis of the parameter values in table 2 and 3?
  7. Kindly capitalize the unit Grays.

Author Response

The authors thank the editor for the opportunity to revise and improve the manuscript, as well as the reviewers for their careful analysis in addition to their insightful suggestions and comments. The concerns of the reviewers are addressed in the current document, while a new version of the manuscript was uploaded with the changes in blue.

Comments 1: The title of the subsection 2.2. is NTCP empirical model, however, the content of the subsection is of LKB model. Kindly crosscheck the content/subsection title.

Response 1: We agree with the reviewer that the content of subsection 2.2. specifically talks about the Lyman-Kutcher-Burman (LKB) model rather than a general empirical NTCP model. Therefore, we have updated the title of subsection 2.2. to “Lyman-Kutcher-Burman model” to reflect the content more accurately. This correction can be found on page 3, Section 2.2., line 96.

Comments 2: The title of the subsection 3.1. is Tumor growth model. However, it is essential to mention if it is in vivo, in vitro, or in silico model.

Response 2: We agree with the reviewer that specifying the nature of the model improves clarity. As the tumor growth model presented in this work is entirely computational, we have updated the title of subsection 3.1. to “In silico tumor growth model.” This change is reflected on page 3, Section 3.1., line 104.

Comments 3: Significant grammatical modification is needed in line 98-100

Response 3: We changed the formulation. “The environment formulation is the same as in the previous work done by Moreau et al. [4]. The model representation is made up of four 50 x 50 2D grids, as shown in Fig. 1, where each pixel on the grid corresponds to: The model representation is made up of four 50 x 50 2D grids, as represented in Fig. 1, where each pixel on the grid corresponds to:”, see page 3, section 3.1 lines 105-107.

Comments 4: What does two numbers (3.6 × 10−9, 1.2 × 10−9) indicate in line 124 and several others?

Response 4: We agree that the notation was not sufficiently explicit. We have revised the text to clarify that the values are the parameters of normal distributions: the first indicates the mean, and the second the standard deviation. Additionally, we now specify that at initialization, each cell is assigned a metabolic efficiency parameter sampled from these distributions. This clarification has been added to page 4, section 3.1, lines 123-134.

Comments 5: Everywhere else, kindly mention the type of tumor growth model.

Response 5: We have revised the manuscript to explicitly refer to the tumor growth model as the “in silico tumor growth model”. This clarification has been applied consistently throughout the manuscript to improve precision and readability. At Page 2 section 2.1, lines 82-85, we added: “In subsequent sections of this article, we will use the term "tumor growth model" to refer as our "in silico agent-based model" to avoid redundancy of the in silico term and prevent any confusion with the term "agent" as used in the context of RL.”

Comments 6: What is the basis of the parameter values in table 2 and 3?

Response 6: We have clarified in the revised manuscript the methodology used to determine the parameter values presented in Tables 2 and 3.

For Table 2, the α and β parameters for normal tissues were directly taken from Kehwar et al.  [24], which references the work of Enami et al.  [3]. These values reflect the radio-sensitivity of late-responding tissues, typically associated with organs at risk (OARs), and are consistent with an α/β ratio of approximately 3. For tumor tissues, we based our approach on the α/β ratio reported in van Leeuwen et al.  [26], which is representative of early-responding tissues (with α/β ≈ 10). We selected α-β pairs that satisfy this ratio and ensure complete tumor eradication under conventional treatment protocols, such as 50.4 Gy in 28 fractions. To support this, we generated a range of α values and determined the corresponding β values and make a linear fit based on the prescribed total dose. This procedure allowed us to capture various clinically realistic tumor responses across the three different sites and different total prescribed dose. The revised explanation can now be found at page 4-5 in Section 3.2, lines 162-168.

For Table 3, the parameters TD₅₀ and m are mean values based on a meta-analysis by Dennstädt et al. [25], for each endpoint (rectal bleeding, xerostomia, and pneumonitis). While n is intentionally chosen a much smaller value than the literature to show a lower bound for the NTCP. These choices are now more explicitly stated in the manuscript. The revised version can now be found at page 6 in Section 3.4, lines 214-218. A paragraph has been added to the discussion to underline the importance of the workflow used in the paper. You can see it at page 11-12, Section 6.2., lines 386-390.

For a sake of space, one small paragraph was removed from the manuscript (page 6 section 3.4). “For our analysis, we focused on "rectal bleeding of grade 2 or greater." Xerostomia refers to a reduction in stimulated salivary flow post-treatment. The endpoint used was "25% or less saliva 12 months after radiation therapy." Pneumonitis, or lung inflammation, was grouped under various endpoints, such as "symptomatic pneumonitis" or "pneumonitis of grade 2 or greater" (CTCAE, RTOG, and Southwest Oncology Group), following Dennstadt et al. [25].”

Comments 7: Kindly capitalize the unit Grays.

Response 7: Thank you for pointing out this mistake, we modified this in the new version of the manuscript at page 7 section 4.2 line 263.

Reviewer 2 Report

Comments and Suggestions for Authors

The manuscript integrates reinforcement learning (RL) with a tumor growth model to optimize fractionated radiotherapy treatment schedules, and I liked the concept a lot. However, I have few suggestions before I accept the manuscript. 

@ The authors need to further add in their discussion regarding the role of specific signaling pathways in cancer progression.

@The manuscript should include an overall flow diagram that visually showcases the proposed framework or conceptual model. 

@ The authors should consider splitting the "Limitations and Future Work" section into two distinct sections. The "Limitations" section should focus specifically on the constraints and potential weaknesses of the study, while the "Future Work" section should provide clear, actionable suggestions for how future research could address these limitations and further advance the field. 

@ The manuscript should include a clear and concise list of contributions to the field, outlining how the study advances current knowledge. Additionally, after discussing each concept in the literature review, the authors should explicitly identify the research gap that their study addresses. This will help clarify the novelty and significance of the work in relation to existing studies and provide a more focused context for the contributions of the manuscript.

@ The manuscript should emphasize how the results could be applied in clinical settings or real-world scenarios. 

@ A separate subsection should be added to discuss the dataset details comprehensively. This section should include information such as the source of the dataset, its characteristics (e.g., sample size, type of data), preprocessing steps, inclusion/exclusion criteria, and any ethical considerations.

@ To enhance the depth of the introduction section and broaden the scope of the literature review, we strongly recommend incorporating recent references that are highly relevant to the themes of tumor progression, immune modulation, and advanced cancer therapies. I got some recent related references from the google that can be added and suggest authors as well to carry out a short survey to include recent literature that not only cover key aspects of molecular mechanisms but also highlight cutting-edge therapeutic strategies in oncology. Authors can add these if they want or can even search for the new ones but not older than 2023. 

  • 10.3390/ph16010037

  • 10.1038/s42003-024-06488-9

  • 10.1007/s12035-023-03911-w

  • 10.3892/or.2024.8776

  • 10.1186/s13046-025-03297-8

  • 10.1021/acsami.4c00793

@ Please make the conclusion broader. 

Author Response

The authors thank the editor for the opportunity to revise and improve the manuscript, as well as the reviewers for their careful analysis in addition to their insightful suggestions and comments. The concerns of the reviewers are addressed in the current document, while a new version of the manuscript was uploaded.

Comments 1: The authors need to further add in their discussion regarding the role of specific signaling pathways in cancer progression.

Response 1: We agree that specific signaling pathways, such as those involved in angiogenesis, apoptosis, or DNA repair, play a crucial role in cancer progression and therapeutic response. While our model does not explicitly simulate molecular signaling cascades, it incorporates the effect of angiogenesis and DNA repair in a simplified manner, through stochastic parameters that influence oxygen and glucose availability across the tumor grid for example. This allows us to capture the spatial heterogeneity resulting from variable vascular support. We have added a sentence in the discussion (page 13 Section 6.5 lines 469-475) to clarify this modeling choice and to highlight the potential of integrating more detailed pathway-level dynamics in future work. Moreover, we added one paper in the Introduction section talking about a cancer digital twin taking into account more signaling networks [9] at page 1, Section 1, lines 29-34.

Comments 2: The manuscript should include an overall flow diagram that visually showcases the proposed framework or conceptual model.

Response 2: In response, we added a new figure (now Figure 1) that presents a comprehensive flow diagram of the proposed framework. This new diagram provides a clearer overview of the entire reinforcement learning process, including the MDP formulation, the interaction between the agent and the tumor environment, and the integration of the NTCP model. As a result, the previous Figure 2 and Figure 1, which was more generic and focused only on the MDP concept, has been removed to both avoid redundancy and to comply with the page limit. We believe the new version improves the clarity and completeness of the manuscript. The new figure (Figure 1) is visible at page 3 of the manuscript, section 3.1.

Comments 3: The authors should consider splitting the "Limitations and Future Work" section into two distinct sections. The "Limitations" section should focus specifically on the constraints and potential weaknesses of the study, while the "Future Work" section should provide clear, actionable suggestions for how future research could address these limitations and further advance the field.

Response 3: We fully agree with the reviewer’s recommendation. As advised, we have split the original "Limitations and Future Work" section into two distinct subsections: Section 6.4 now presents the study’s limitations, while Section 6.5 outlines potential directions for future work (page 12-13, lines 422-482). This change improves the clarity of the discussion and better distinguishes between the current constraints of our approach and proposed extensions for future research.

Comments 4: The manuscript should include a clear and concise list of contributions to the field, outlining how the study advances current knowledge. Additionally, after discussing each concept in the literature review, the authors should explicitly identify the research gap that their study addresses. This will help clarify the novelty and significance of the work in relation to existing studies and provide a more focused context for the contributions of the manuscript.

Response 4: In response, we have revised the Introduction section to explicitly delineate the research gap and our study's contributions. While previous studies have explored reinforcement learning (RL) approaches for radiotherapy dose fractionation, there is a lack of comprehensive integration of RL with Normal Tissue Complication Probability (NTCP) models to evaluate treatment plans. Specifically, existing research often does not compare RL-based treatment schedules with conventional fractionation schemes using NTCP assessments for specific organs and complications.

Our Contributions:

  1. We develop an RL-based framework that determines daily dosage in hypofractionated radiotherapy treatment planning, integrating a 2D tumor growth model with Dose-Volume Histogram (DVH) analysis and the Lyman NTCP model.
  2. We provide a comparative analysis between RL-based treatments and traditional counterparts, focusing on total administered dose, number of fractions required for tumor eradication, and the likelihood of complication occurrences for specific organ locations.
  3. We evaluate the robustness and adaptability of our RL agents when confronted with uncertainties in the tumor growth model, ensuring efficacy across a broad spectrum of model parameters.

We have updated the manuscript accordingly to reflect these clarifications and enhancements but we're leaving this part of the manuscript in text format instead of putting it in list format. This can be seen at page 2 section 1 lines 45-53.

Comments 5: The manuscript should emphasize how the results could be applied in clinical settings or real-world scenarios.

Response 5: To address the reviewer's comment regarding the applicability of our results in clinical settings, we have expanded the "Future Work" section to discuss potential ways for translating our in silico findings into real-world clinical practice (page 13 section 6.5 lines 457-468).

Comments 6: A separate subsection should be added to discuss the dataset details comprehensively. This section should include information such as the source of the dataset, its characteristics (e.g., sample size, type of data), preprocessing steps, inclusion/exclusion criteria, and any ethical considerations.

Response 6: We would like to clarify that our study is entirely based on an in silico framework and does not rely on any external or clinical dataset. Instead, the tumor microenvironment and response to treatment are simulated in real time through a mechanistic agent-based model, whose parameters are derived from previously published literature. The reinforcement learning agent interacts directly with this simulated environment, and transitions are computed on-the-fly during training without storing an external dataset. Therefore, a dataset description section is not applicable in this context. However, to prevent future confusion, we have clarified this point explicitly in the revised manuscript (see page 4, Section 3.1, lines 145-150).

Comments 7: To enhance the depth of the introduction section and broaden the scope of the literature review, we strongly recommend incorporating recent references that are highly relevant to the themes of tumor progression, immune modulation, and advanced cancer therapies. I got some recent related references from the google that can be added and suggest authors as well to carry out a short survey to include recent literature that not only cover key aspects of molecular mechanisms but also highlight cutting-edge therapeutic strategies in oncology. Authors can add these if they want or can even search for the new ones but not older than 2023. 

    1. 10.3390/ph16010037
    2. 10.1038/s42003-024-06488-9
    3. 10.1007/s12035-023-03911-w
    4. 10.3892/or.2024.8776
    5. 10.1186/s13046-025-03297-8
    6. 10.1021/acsami.4c00793

Response 7: We have added several more recent references about 3D models, radiation-response model and Cell Kinetics with Metabolism, Signaling Networks, and Biomechanics integration in a model. Those were added at page 1, in Section1, lines 29-34.

  • Bulletin of Mathematical Biology (2024) 86:139, https://doi.org/10.1007/s11538-024-01371-4, AMBER: A Modular Model for Tumor Growth, Vasculature and Radiation Response Louis V. Kunz · Jesús J. Bosque · Mohammad Nikmaneshi · Ibrahim Chamseddine · Lance L. Munn · Jan Schuemann · Harald Paganetti · Alejandro Bertolet, October 2024
  • Developing an Agent-Based Mathematical Model for Simulating Post-Irradiation Cellular Response: A Crucial Component of a Digital Twin Framework for Personalized Radiation Treatment Ruirui Liu, Marciek H. Swat, James A Glazier, Yu Lei, Sumin Zhou, Kathryn A. Higley, 21 Jan 2025,
  • Mathematical modeling in radiotherapy for cancer: a comprehensive narrative review Dandan Zheng1*, Kiersten Preuss, Michael T. Milano, Xiuxiu He, Lang Gou, Yu Shi, Brian Marples, Raphael Wan, Hongfeng Yu, Huijing Du and Chi Zhang, Zheng et al. Radiation Oncology (2025) 20:49, https://doi.org/10.1186/s13014-025-02626-7
  • A Multidisciplinary Hyper-Modeling Scheme in Personalized In Silico Oncology: Coupling Cell Kinetics with Metabolism, Signaling Networks, and Biomechanics as Plug-In Component Models of a Cancer Digital Twin, by Eleni Kolokotroni, Daniel Abler, Alokendra Ghosh, Eleftheria Tzamali, Journal of Personalized Medicine, Vol. 14, Issue 5, 3390/jpm14050475

The comment of the reviewer allows us to add a limitation to our paper. We thus added two references suggested by the reviewer (10.3390/ph16010037 or 10.1021/acsami.4c00793) in the Limitation section (page 13 Section 6.5 lines 475-482). All the modifications in the references can be found at page 15 lines 549-557 and 593-597.

Comments 8: Please make the conclusion broader. 

Response 8: In response, we have revised the conclusion to provide a more comprehensive perspective on the study's implications and future directions. Specifically, we have elaborated on the clinical relevance of our findings, discussing how reinforcement learning (RL) could be integrated into real-world radiotherapy treatment planning. We have also highlighted observed trends in dose scheduling, noting the tendency for higher doses at the beginning of treatment sessions, followed by gradual reductions, and discussed the potential clinical significance of this pattern as well as reduction of NTCP with our RL method. Furthermore, we have proposed future enhancements to our model, including the transition to three-dimensional simulations and the incorporation of medical imaging data, to better reflect clinical scenarios. These additions aim to provide a more holistic overview of our study's contributions and its potential impact on the field of radiotherapy. This can be seen at page 14, Section 7, lines 498-511.

Reviewer 3 Report

Comments and Suggestions for Authors

The manuscript titled "Optimal fractionation scheduling for radiotherapy treatments with reinforcement learning (RL), tumor growth modeling and outcome modeling" addresses a potentially interesting topic in radiobiology and cancer therapy in general. The results demonstrate that the RL-based approach in radiotherapy not only achieves tumor eradication but also significantly reduces healthy tissue damage compared to traditional treatment methods. The study is novel, well-designed, and the manuscript is well-written. Following are few minor issues need to be addressed:

  1. Lines 54–55: The information currently presented in the Introduction would be more appropriately placed in the Data Availability Statement and/or the Methods section.
  2. Figure 1: Please label all panels clearly (e.g., (a), (b), (c), etc.) within the figure itself. In addition, the figure legend should provide explicit descriptions for each panel, including definitions of all color codes and their corresponding cell types or experimental conditions.
  3. The limitation of the 3D tumor model should be addressed in discussion.
  4. The manuscript would benefit from a thorough language revision. Please carefully revise the grammar, sentence structure, and formatting throughout to improve clarity and readability.
Comments on the Quality of English Language
  1. The manuscript would benefit from a thorough language revision. Please carefully revise the grammar, sentence structure, and formatting throughout to improve clarity and readability.

Author Response

The authors thank the editor for the opportunity to revise and improve the manuscript, as well as the reviewers for their careful analysis in addition to their insightful suggestions and comments. The concerns of the reviewers are addressed in the current document, while a new version of the manuscript was uploaded.

Comments 1: Lines 54–55: The information currently presented in the Introduction would be more appropriately placed in the Data Availability Statement and/or the Methods section.

Response 1: We agree with the reviewer’s comment. As suggested, we have moved this sentence from the Introduction to the Methods section, where it now appears on page 7, Section 4, lines 235-236. The same information was already included in the Data Availability Statement, and we have chosen to retain it there as well for completeness (lines 529-530).

Comments 2: Figure 1: Please label all panels clearly (e.g., (a), (b), (c), etc.) within the figure itself. In addition, the figure legend should provide explicit descriptions for each panel, including definitions of all color codes and their corresponding cell types or experimental conditions.

Response 2: Following the comment of Reviewer 2, we merge information of previous Figure 1 and 2 and we add a new figure (now Figure 1) representing the general workflow of this paper. The figure concerned by this comment is thus not present anymore in the manuscript, but I added a legend (color code) for the different cell types in the representative matrix in Figure 1 (page 3, Section 3.1).

Comments 3: The limitation of the 3D tumor model should be addressed in discussion.

Response 3: We have to clarify that the current model is two-dimensional, and not three-dimensional as you thought. We have revised the manuscript to clarify this point and to explicitly address the limitations associated with using this 2D tumor growth model. This has now been discussed in Section 6.4 (“Limitations”) at page 12-13, Section 6.4., lines 438-441. In particular, we note that a 2D model simplifies tumor architecture, spatial interactions, and nutrient gradients, and does not capture volumetric effects seen in realistic 3D environments. We highlight these points as goals for future improvements. You can see this at page 13, Section 6.5., lines 469-475.

Comments 4: The manuscript would benefit from a thorough language revision. Please carefully revise the grammar, sentence structure, and formatting throughout to improve clarity and readability.

Response 4: We have carefully revised the manuscript to improve the grammar, sentence structure, and overall readability. The revised version has been reviewed by a native English speaker with experience in scientific editing. While we did not highlight every language-related change directly in the manuscript, all modifications have been made consistently throughout the text. We confirm that none of the scientific content or intended meaning has been altered as a result of this revision.

Round 2

Reviewer 2 Report

Comments and Suggestions for Authors

The comments are addressed.